# Role of Climate and Edaphic Factors on the Community Composition of Biocrusts Along an Elevation Gradient in the High Arctic

**DOI:** 10.3390/microorganisms12122606

**Published:** 2024-12-17

**Authors:** Isabel Mas Martinez, Ekaterina Pushkareva, Leonie Agnes Keilholz, Karl-Heinz Linne von Berg, Ulf Karsten, Sandra Kammann, Burkhard Becker

**Affiliations:** 1Department of Biology, Institute for Plant Sciences, University of Cologne, 50674 Cologne, Germany; imasmart@smail.uni-koeln.de (I.M.M.); ekaterina.pushkareva@uni-koeln.de (E.P.); lkeilho1@uni-koeln.de (L.A.K.); linnevonberg@uni-koeln.de (K.-H.L.v.B.); 2Institute for Biological Sciences, University of Rostock, 18059 Rostock, Germany; ulf.karsten@uni-rostock.de (U.K.);

**Keywords:** biological soil crusts, cyanobacteria, micro algae, climate, edaphic factors

## Abstract

Biological soil crusts are integral to Arctic ecosystems, playing a crucial role in primary production, nitrogen fixation and nutrient cycling, as well as maintaining soil stability. However, the composition and complex relationships between the diverse organisms within these biocrusts are not well studied. This study investigates how the microbial community composition within Arctic biocrusts is influenced by environmental factors along an altitudinal gradient (101 m to 314 m). Metagenomic analyses were used to provide insights into the community composition, revealing that temperature, pH, and nutrient availability significantly shaped the community. In contrast, altitude did not directly influence the microbial composition significantly. Eukaryotic communities were dominated by Chloroplastida and fungi, while Proteobacteria and Actinobacteria prevailed among prokaryotes. Cyanobacteria, particularly orders such as Pseudoanabaenales, Pleurocapsales, and Nostocales, emerged as the most abundant photoautotrophic organisms. Our findings highlight the impact of environmental gradients on microbial diversity and the functional dynamics of biocrusts, emphasizing their critical role in Arctic tundra ecosystems. Arctic biocrusts are intricate micro-ecosystems, whose structure is strongly shaped by local physicochemical parameters, likely affecting essential ecological functions.

## 1. Introduction

The Arctic region is characterised by harsh environmental conditions, including low temperatures, short growing seasons and limited water availability, which hinder the growth of numerous organisms. Despite these challenges, biological soil crusts (biocrusts) constitute a substantial proportion of the vegetation in Arctic ecosystems [1]. Biocrusts are communities of autotrophic and heterotrophic organisms, including bacteria, cyanobacteria, microalgae, lichens, and bryophytes that exist in mutual association [2]. They form a thin layer covering the soil surface, typically 5–10 mm thick, and constitute a significant proportion of the total vegetation in Arctic regions. Consequently, biocrusts play a crucial role in primary production and provide a number of benefits including soil surface stabilisation, nitrogen fixation and organic matter accumulation [2,3].

The microbial community composition of biocrusts in Arctic regions encompasses a diverse array of species with distinct ecological functions, shaped by environmental and soil conditions. However, the dominant phyla are often consistent, with typical bacterial phyla including Actinobacteria, Acidobacteria, cyanobacteria and Proteobacteria [4]. Photoautotrophic phyla such as eukaryotic green algae, bryophytes, tracheophytes and cyanobacteria play an important role in such communities, driving primary production within biocrusts [1,3]. The algal classes most commonly found in biocrusts include Klebsormidiophyceae, Zygnematophyceae, Chlorophyceae, Trebouxiophyceae and Ulvophyceae, while the most frequently occurring cyanobacterial orders are Nostocales, Oscillatoriales and Pseudanabaenales [5,6]. In Arctic regions, the combination of low temperatures, low soil moisture due to minimal precipitation, and limited soil nutrients restricts primary production [7]. Lichens, symbiotic associations of fungi and algae (e.g., *Trebouxia* sp.), are well adapted to these harsh conditions and become more prominent as the climate becomes drier [8]. In addition, nitrogen fixation by bacteria is a critical source of nitrogen in Arctic ecosystems. For example, heterocystous cyanobacteria contribute significantly to soil nitrogen fixation [9].

Svalbard is a Norwegian archipelago located in the Arctic Ocean. Despite its location between continental Norway and the North Pole, the climate is moderated by the North Atlantic Current. The mean temperature in Svalbard remains below 10 °C throughout the summer, with the warmest months occurring between the end of May and the end of August. The winter months are characterised by a decline in temperature to an average of −10 °C. The precipitation level rarely exceeds 50–60 mm per month [10]. Furthermore, Svalbard’s northern geographic location results in extreme variations in daylight throughout the year, with approximately 90 days of polar night in winter, characterised by continuous darkness, and 90 days of midnight sun in summer, during which the archipelago experiences uninterrupted sunlight [11].

The flora of Svalbard is diverse, comprising approximately 160 species of vascular plants, 380 mosses and 600 lichens [12]. The vegetation is predominantly characterised by low-growing species that form dense ground covers, with a variety of plants coexisting in close proximity.

The objective of this study is to enhance our understanding of the microbial community composition of Arctic biocrusts and the climatic and edaphic factors shaping these communities. To achieve this, an altitudinal gradient (101–314 m a.s.l.) was selected as the basis for this study, enabling the investigation of biocrust responses to varying environmental conditions. This gradient reflects the diverse landscape and topography surrounding Kongsfjorden. Kern et al. [13] previously documented significant environmental variation, including soil nutrient levels and water availability, across small spatial scales along the toposequences of Ossian-Sarsfjellet. This study primarily investigates the community composition of photoautotrophic organisms, providing new insights into how microbial communities respond to changing environmental conditions, particularly in extreme Arctic environments. Given the typical changes in environmental conditions with increasing altitude, we hypothesised that harsher environmental conditions at higher altitudes would result in a decline in larger vascular plants, resulting in distinct shifts in microbial community composition. Furthermore, we anticipated that variations in temperature, soil moisture and soil chemistry along the altitudinal gradient would be reflected in the composition of microbial communities.

## 2. Materials and Methods

### 2.1. Site Description and Sampling

The sampling sites were located on the slope of the larger mountain on Ossian Sarsfjellet in Svalbard (see Table 2 for GPS coordinates). Four sampling sites, situated at altitudes ranging from 101 m to 314 m a.s.l., were selected for this study at different altitudes (Figure 1). Site 1 was located near Sarsvatnet lake, while Sites 2, 3 and 4 were positioned on the eastern slope of the larger mountain on Ossian Sarsfjellet. The sites were selected in the interior of the island to minimise the influence of the adjacent Kongsfjorden to the west and the glaciers to the east. The soil remained free of snow throughout the project.

Three TOMST^®^ data loggers (TOMST S.R.O., Prag, Czech Republic) were placed in the soil at each site to measure temperature and soil moisture. The temperature measurements were obtained from each sensor at three different positions: 6 cm below the soil surface (T1), the soil surface (T2) and atmospheric temperature 15 cm above the soil surface (T3). Measurements were recorded at 15 min intervals from 13 July 2023 to 21 August 2023. One sensor at Site 1 and all three sensors at Site 2 were removed from the ground due to external factors (most likely animals). Consequently, the data from these covered the period only from 13 July to 4 August 2023. Additionally, data from the meteorological station in Ny-Ålesund (station ID: SN99910) were also used for further analyses.

The minimum fluorescence of the opened reaction centres of photosystem II (PSII) (F0, basal fluorescence of chlorophyll *a*) and the maximum fluorescence (Fm) were measured with a hand-held field PAM (FluorPen 100 instrument; Photon Systems Instruments, Drásov, Czech Republic) using the OJIP protocol with the following settings: flash pulse 20% and super pulse 80%, after 15 min of dark acclimation. The maximum quantum yield of PSII (QYmax = (Fm − F0)/Fm) was calculated.

Three replicates of the biocrust material were collected at each site on 13 July 2023.

### 2.2. Vegetation Analysis

The Braun-Blanquet vegetation analysis was conducted on Ossian Sarsfjellet, Svalbard, on 13 July 2023. A total of 12 plots were analysed, with three individual plots of 1 m^2^ selected at each site. For each plot, individual species were recorded, along with the general layout and appearance of the site. In addition, the coverage of each species was estimated using the scale presented in Table 1. Statistical analyses were conducted to compare the vegetation at different altitudes and identify significant differences in species composition. For statistical purposes, the Braun-Blanquet vegetation scale was converted into numerical values (Table 1).

### 2.3. Soil Analysis

The chemical analysis of the soil was conducted at the University of Rostock. Samples were dried at 45 °C for 24 h prior to analysis. Total carbon (TC) and total nitrogen (TN) were measured using the Vario EL CNS analyser (Elementar Analysensysteme GmbH, Hanau, Germany). Total phosphorus (TP) was measured following microwave-assisted digestion with aqua regia solution. The concentration was then quantified by inductively coupled plasma (ICP) optical emission spectroscopy using the Optima 8300 ICP-OES Optical System (PerkinElmer, Waltham, MA, USA) at a wavelength of 214 mm.

Soil pH was determined electrometrically in a 0.01 M calcium chloride (CaCl_2_) solution with a weight/volume ratio (*w*/*v*) of 1:2.5, according to DIN ISO 10390 [14]. Measurements were taken after 30 min using a pH meter (METTLER TOLEDO SevenMulti, Giessen, Germany).

The chlorophyll *a* content was measured by adding 0.1 g of magnesium carbonate (MgCO_3_) to each sample, and then extracting chlorophyll *a* in 3 mL of 96% ethanol at 78 °C for 30 min. The concentration was measured using a spectrophotometer (Shimadzu UV-2401 PC, Kyoto, Japan) at wavelengths of 750, 696, 665, 649 and 632 nm. The final chlorophyll *a* content was calculated according to the given Equation (1).
(1)$$Chl\;a=0.0604\times \left(A_{632nm}-A_{750nm}\right)-4.5224\times \left(A_{649nm}-A_{750nm}\right)\;\;\;\;\;\;\;\;\;\;\;\;\;+13.2969\times \left(A_{665nm}-A_{750nm}\right)-1.7453\times (A_{696nm}-A_{750nm})\;$$

### 2.4. DNA Isolation and Sequencing

DNA was extracted from the collected biocrusts using the DNeasy Power Soil Pro Kit (QIAGEN GmbH, Hilden, Germany) according to the manufacturer’s instructions. The DNAs were further sent to the Cologne Center for Genomics (University of Cologne, Cologne, Germany), where the quality control was conducted on an Agilent 2200 TapeStation System (Agilent Technologies, Santa Clara, CA, USA) with the help of TapeStation Analysis Software 5.1. Metagenomic sequencing was conducted using the Illumina NovaSeq6000 sequencing system (PE150) (Illumina, San Diego, CA, USA). The raw reads were deposited in the Sequence Read Archive (SRA) under the project number PRJNA1189472.

### 2.5. Bioinformatic and Statistical Analyses

The bioinformatic analysis for metagenomics was primarily conducted using the OmicsBox software (Version 3.1.11). FASTQ quality filtering was performed using Trimmomatic (Version 0.38) [15] and rRNA sequences were further extracted using SortMe-RNA [16]. The taxonomic classification of 16S and 18S rRNAs was performed using the Silva database (Version 1.9.10), with default settings and an adjusted cluster sequence identity of 100%. Additionally, the Kraken2 software (Version 2.1.3) was used with a Kraken confidence filter of 0.5 and a minimum hit group of 2 [17].

Statistical analyses were performed in RStudio (Version 4.1.3). Prior to testing, Shapiro–Wilk’s test was performed to assess the normality within the dataset. If necessary, the data were square root transformed. Differences between the individual sampling sites and further parameters were tested with one- and two-way analysis of variance (ANOVA), followed by Tukey’s HSD, with a significance threshold of *p* < 0.05. To visualise the distribution of reads at the different sites, a Non-Metric Multidimensional Scaling (NMDS) plot was generated, using the *vegan* package (Version 2.6−6.1) [18]. Additionally, the *envfit* function was used to fit parameters such as temperature, moisture and soil chemistry. Statistical significance was tested using ANOSIM.

For the vegetation analysis, PERMANOVA (Permutational Multivariate Analysis of Variance) was used to assess differences between the individual sampling sites (*p* < 0.05). Statistical dispersion was checked to validate the PERMANOVA results, followed by ANOSIM for confirmation.

### 2.6. Calculation of the Average Dew Point

For the calculation of the average dew point, the following equations were used. Initially, the saturation vapor pressure (e_s_) was calculated using the Magnus–Tetens Formula (2); here, T denotes the daily average temperature [°C], measured 15 cm above the soil surface through TOMST^®^ data loggers.
(2)$$e_{s}=6.112\times e^{\left(\frac{17.67\times T}{T+243.5}\right)}$$

Following this, the actual vapor pressure (e) was calculated (3); RH signifies the daily relative humidity [%], measured at Ny-Ålesund station (Station ID: SN99910).
(3)$$e=\frac{RH}{100}\times e_{s}$$

Finally, the dew point temperature (Td) was calculated (4), using the previously calculated actual vapor pressure (e).
(4)$$Td=\frac{243.5\times ln(\frac{e}{6.112})\;}{17.67-ln(\frac{e}{6.112})\;}$$

## 3. Results

### 3.1. Environmental Parameters

The data from the weather station in Ny-Ålesund (station ID: SN99910) were used for comparison and to provide an overall picture of the meteorological conditions. Temperature and soil moisture varied between the different sites. Site 2, located at 186 m a.s.l., exhibited the highest temperatures (Figure 2). In contrast, Site 4, situated at 314 m a.s.l., demonstrated the lowest temperatures at all three measurement points (T1, T2, T3). The temperature values recorded above the soil surface exhibited the most pronounced fluctuations throughout the measurement period.

In July and August 2023, winds predominantly originated from the east–southeast direction, with an average speed of approximately 5–6 m/s. Between 15 July and 16 July 2023, weak winds were recorded (3 m/s), which intensified on 17 July 2023, reaching speeds of 9 m/s (Appendix A). Considering the geographical location of the four sites, Site 4 experienced the highest exposure, particularly due to its elevated position on the larger Ossian Sarsfjellet mountain.

The highest soil moisture values were observed at Site 4, while the lowest values were recorded at Site 2 (Figure 3). A sudden increase in soil moisture was observed at the beginning of August. A comparison with meteorological data from the weather station in Ny-Ålesund indicates that the observed increase in soil moisture was due to rainfall events during this period. Following the rainfall, the soil moisture levels recorded at Sites 3 and 4 remained elevated throughout August.

The dew point was calculated using the formulas explained in the Section 2 (Formulas (2)–(4)) to explain the highest soil moisture observed at Site 4. It can be assumed that on days when the temperature declines below the calculated dew point, the formation of dew at the location is likely. Consequently, Site 4 had the most days with possible dew formation. Conversely, Site 2 demonstrated the least degree of overlap, in accordance with the measured soil moisture data.

The soil pH of the biocrusts was neutral to slightly acidic at all sites, ranging from 6.13 to 7.23 (Table 2). Significant differences were observed in the pH levels between Sites 1 and 3, which exhibited a pH of approximately 6, and Sites 2 and 4, which demonstrated a pH of approximately 7. Site 3 had the lowest total phosphorus (TP), total carbon (TC) and total nitrogen (TN) contents and the highest carbon to nitrogen ratio (C/N) of 18.6. The chlorophyll *a* content of the biocrusts did not show any significant changes with increasing elevation (Table 2). Similarly, the quantum yield capacity of photosystem II, (Fv/Fm) measured with a field PAM, remained unchanged (Table 2).

### 3.2. Vegetation at the Sites

The vegetation at all of the sampling sites was predominantly composed of small vascular plants, mosses, lichens and biocrusts. A total of 26 different species were recorded (Appendix A). Of these, 12 species could be assigned to eight different families of tracheophytes, ten different species of bryophytes and four different species of lichens. A total of 3 of the 26 species could not be identified. A decline in the total number of species per site was observed, with Site 1 exhibiting the highest number of species (17) and Site 4 the lowest (15). Furthermore, the overall ground cover of the species demonstrated a tendency to decrease with increasing altitude.

The highest ground cover was observed for *Dryas octopetala*, *Sterecaulon alpinum*, *Salix polaris*, *Carex* sp., *Polygonum viviparum* and one unidentified lichen species (Appendix A). *Dryas octopetala* exhibited the highest ground cover, reaching up to 50% at Site 2. While species such as *Dryas octopetala*, *Sterecaulon alpinum*, *Salix polaris* and *Saxifraga oppositifolia* were present at all four sites with relatively high cover ratios, *Silene acaulis* was exclusive to Site 1 and the two bryophytes *Thamnolia vermicularis* cf. and *Dicranella* cf. were only recorded at Site 4. The abundance of Tracheophyta exhibited a decline in abundance with increasing altitude (Appendix A), whereas lichen increased with higher sites. In contrast, the distribution of bryophytes did not show a clear gradient. Instead, similar abundance patterns were observed between Sites 1 and 3, as well as between Sites 2 and 4. NMDS analysis, along with PERMANOVA and ANOSIM significance tests, revealed that the four sites represented different communities. The most important abiotic factors for explaining these differences were temperature, pH, moisture and nutrient contents (Figure 4).

### 3.3. Sequencing Overview

Metagenomic sequencing generated around 13.5 million initial reads per sample. Following quality filtering, 2.3% were removed (Appendix A). Approximately 0.2% of the quality-filtered reads consisted of rRNAs, which were used for taxonomic composition analyses.

### 3.4. Metagenomic Profile of the Biocrusts

Taxonomic classification of the 16S and 18S rRNA reads using the Silva database revealed a diverse range of taxa, with 89.2% of reads belonging to bacteria, 6.6% to eukaryota, 0.1% to archaea and 3.8% of reads unclassified (Figure 5). The highest proportion of bacterial reads was observed in Site 2 (93%), while Site 3 had the lowest number of bacterial reads (87%). The dominant bacterial phyla were Actinobacteria and Proteobacteria. Actinobacteria dominated Site 1 (23.5% of total reads) and Site 2 (29% of total reads), while Proteobacteria were the most prevalent at Site 3 (22.6% of total reads) and Site 4 (26.7% of total reads). Fungi were the dominant eukaryotic phyla at Sites 2, 3 and 4 (1.2, 4 and 3.1% of total reads, respectively), while Chloroplastida dominated Site 1 (3.4% of total reads).

The NMDS analysis and ANOSIM significance test revealed significant similarities between the sites (Appendix A), with Sites 1 and 3 displaying the most pronounced similarities. The observed similarities were mainly driven by environmental parameters, particularly temperature, soil moisture, pH, total nitrogen and total phosphorus, and C/N ratio.

The analysis of fungal rRNA reads revealed a clear dominance of Ascomycota, accounting for up to 83% of fungal reads at Site 3 (Figure 6A). Basidiomycota represented an average of 12% of fungal reads, while Mortierellomycota contributed to approximately 9% of fungal reads. Functional guild analysis revealed a prevalence of saprotrophic (30% of fungal reads at Sites 3 and 4) and lichenised fungi (23% of fungal reads at Site 2) (Figure 6B). Additionally, 14 indicator taxa were identified (Appendix A). Nine of these taxa were recorded at Site 3, belonging to the Ascomycota (*Epiglia*, *Sporoschismopsis*, *Rhytisma*, *Ascosphaera*, *Leotia*, *Ochrolechia*, *Umbilicaria*) and the Mucoromycota (*Mortierella*). Two indicator taxa from the Basidiomycota and two indicator taxa from the Ascomycota were identified at Sites 1, and 4, respectively (Appendix A).

The sequences for photoautotrophic taxa, extracted from the 16S and 18S datasets, showed that approximately 6% of all organisms recorded at the sites were classified as primary producers. Cyanobacteria were the most prevalent group at all sites, representing up to 84% of photoautotrophic reads, and exhibited no significant correlation between abundance and altitude (Figure 7). In contrast, the majority of algal groups, bryophytes and tracheophytes demonstrated a pattern of declining abundance at higher elevations. Site 2 exhibited a comparatively low abundance of eukaryotic photoautotrophic species (Figure 7). However, the green algae family Trebouxiophyceae showed a non-significant increase in abundance with increasing altitude from 4% to 8% of photoautotrophic reads, with the highest abundance observed at Site 3. A total of nine indicator species were identified among all photoautotrophic taxa, of which seven were identified as Cyanobacteria (*Phormidium*, *Hormoscilla*, *Leptolyngbya* and *Loriellopsis*) (Appendix A).

The cyanobacterial species were predominantly composed of Pseudoanabaenales, accounting for up to 40% of the total cyanobacterial population, followed by Nostocales and Gloeobacterales (Figure 8). In general, filamentous species constituted the majority of the cyanobacterial population, with approximately 48% of cyanobacterial reads at Sites 1 and 2, and approximately 44% of cyanobacterial reads at Sites 3 and 4. The unicellular orders, Synechococclaes, Gloebacteriales and Chroococcodiopsiales, exhibited a consistent increase in abundance from Site 1 (21% of cyanobacterial reads) to Site 4 (34% of cyanobacterial reads) (Figure 8). The highest abundance of heterocystous cyanobacteria (Nostocales) was recorded at Site 1 (25.5% of cyanobacterial reads), while the lowest abundance was observed at Sites 2 and 4 (14.7% and 15.1% of cyanobacterial reads) (Figure 8). Furthermore, *Nostoc* sp., *Petalonema* sp. and *Scytonema* sp. exhibited the highest abundances, with averages of 5, 4 and 3% of cyanobacterial reads, respectively, across all sites. Overall, *Gloeobacter* sp., belonging to the order Gloeobacterales, was the most dominant species, with an abundance of up to 23.6% of cyanobacterial reads at Site 3 and approximately 9.7% of cyanobacterial reads at Sites 1, 2 and 4.

## 4. Discussion

### 4.1. Environmental Conditions

The temperature data recorded at the four sites did not demonstrate a consistent decline with increasing altitude. While Site 4 exhibited significantly lower temperatures, the other three sites showed variability, with Site 2 displaying the highest temperatures and Sites 1 and 3 exhibiting intermediate values. Notably, another study conducted in Svalbard reported a similar temperature pattern across an altitudinal gradient during the summer season, while a clear decline in temperature was observed at higher elevations during the winter months [19]. This variation is likely due to geographical and topographical influences at the selected altitudes, suggesting that the altitude effect between Sites 1 and 4 may have been partly overridden by local spatial or seasonal factors. Meteorological data from the Ny-Ålesund station indicated that local wind patterns and intensities probably influenced the recorded temperatures at the different sites. On 16 July 2023, when wind speeds were relatively low, temperatures were higher at all sites. However, as the wind speeds increased on 17 July 2023, there was a notable decline in temperature, particularly at Site 4, which was the most exposed due to its geographical characteristics. The temperature readings above the soil surface (T3) and at the soil surface (T2) showed pronounced daily fluctuations due to the direct exposure to environmental conditions. T3, which recorded the lowest temperatures and the greatest fluctuations, was significantly influenced by wind and solar radiation. In contrast, soil surface temperatures (T2) exhibited a tendency towards higher temperatures, benefiting from the insulation provided by plant cover, solar absorption and reduced wind exposure [20,21]. Temperatures below the surface (T1) demonstrated greater stability, as the vegetation and soil acted as insulation, reducing sudden temperature changes [21,22].

Although it is generally accepted that temperatures decline with increasing altitude, the influence of soil moisture is more dependent on the specific local conditions [23,24,25]. Warmer temperatures promote soil evaporation, while cooler temperatures reduce it, allowing moisture to accumulate [24]. This pattern was evident in our data, with Site 4, the coldest site, exhibiting the highest soil moisture, and Site 2, the warmest site, displaying the lowest. Furthermore, higher elevations typically experience greater precipitation, which contributes to increased soil moisture. However, no clear pattern of rainfall-related moisture accumulation was observed at the study sites. In regions with low precipitation, alternative sources of water, such as the formation of dew, become significant. Dew forms when temperatures fall below the dew point [26], and although the calculated dew points were based on data from the Ny-Ålesund weather station and may not be entirely accurate for the study sites, they were in good agreement with the measured soil moisture values, potentially reinforcing the existing environmental conditions.

The analysis of soil chemistry revealed no clear altitudinal gradient. Instead, similarities were observed between Sites 1 and 3, as well as between Sites 2 and 4. The pH values at all sites ranged from neutral to slightly acidic, which is consistent with the previous studies conducted in Svalbard [19,27]. The nutrient content generally followed an elevational pattern, with carbon and nitrogen levels increasing at higher elevations and a slight decrease in phosphorus content. However, this trend was less apparent at Site 3, which exhibited different nutrient characteristics compared to the other sites [25,28]. The presence of rocks at these higher elevations may act as nitrogen hotspots by intercepting nutrient-rich precipitation [29], explaining the elevated nitrogen levels observed at the rockier, higher elevation sites. Furthermore, the slower microbial decomposition rates observed at higher elevations are attributed to the lower temperatures, which allow for the accumulation of carbon and nitrogen in the soil [28]. The C/N ratio, a common indicator of organic matter decomposition, provided further insight into nutrient dynamics. A C/N ratio of 18, as recorded at Site 3, may indicate some degree of nitrogen limitation and potentially reduced microbial activity [30]. However, since soils with C/N ratios below 20 are generally considered nutrient-sufficient, it is unlikely that severe competition between plants and microbes for nutrients is occurring at these sites [30]. Site 4, with the highest soil moisture and probably the greatest exposure to precipitation and dew formation, might benefit from nutrient inputs via rainwater, with runoff potentially enriching the lower sites over time. However, the unexpectedly low nutrient content at Site 3 challenges this assumption and suggests more complex, unexplored factors influencing nutrient distribution. This variation suggests that additional variables, such as subtle shifts in community composition, geological conditions, plant–microbe interactions or specific soil characteristics, may play an important role in shaping the nutrient profile at this site [13].

### 4.2. Vegetation Reacts to Increased Altitude

The vegetation analysis conducted at the four sites revealed a general reduction in ground cover with increasing altitude, accompanied by clear patterns between species groups, as commonly observed and well documented in mountains [13,31].

The abundance of larger vascular plants (Tracheophyta) exhibited a decline with increasing altitude. The most frequently occurring species at all four sites were *Dryas octopetala*, *Saxifraga oppositifolia* and *Salix polaris*, highlighting their adaptability to different habitats. *D. octopetala* and *S. oppositifolia* often coexist and dominate a variety of environments in Svalbard [32]. *S. polaris*, which is also widely distributed in Svalbard, typically grows close to the ground and forms wide, spreading mats [31]. This growth pattern is typical to many tracheophytes growing in Arctic tundra regions, including *D. octopetala*, *S. oppositifolia*, *Cassipope tetragona*, *Salix reticulata* and *Silene acaulis* [33]. The mat or cushion structures assist the plants in protecting themselves from harsh environmental conditions by trapping heat during cold periods and cooling the air during the summer, thereby creating microclimates [33,34]. Another common adaptation is the growth of creeping or dwarf shrubs, which keep plants low to the ground, reducing their exposure to strong winds [33,35], as seen in *D. octopetala*, *S. polaris* and *S. reticulata.*

Bryophytes were the most abundant at Sites 2 and 4. A similar pattern was observed in the distribution of total nitrogen and total carbon, pH and carbon/nitrogen ratio, as well as temperature and soil moisture data. Previous studies of bryophytes in the Svalbard region have shown that their abundance tends to increase with altitude [36], which is consistent with our results. Although Site 2 may appear to be contradictory, Prestø’s study [36] also indicated that bryophytes are abundant in dry habitats. This is due to their capacity to absorb water and nutrients over their entire surface area, eliminating the need for a complex root system [36]. Given that Site 2 was the driest surveyed site, this may explain the presence of a large number of bryophytes.

The abundance of lichen species increased in abundance with altitude, accompanied by an increase in ground cover. A total of four lichen species were recorded across all sites, but only two could be identified as *Flavocetraria nivalis* and *Sterecaulon alpinum*. In addition to cyanobacteria, eukaryotic algae and bryophytes, lichens represent a significant component of biocrusts [4,5]. However, the identification of lichen species proved to be challenging due to limited field identification at the sites and the limitations of metagenomic taxonomic classification, where lichens are listed under their fungal names, often providing insufficient information at the species-level [37,38].

### 4.3. Microbial Community Composition in Arctic Biocrusts

The eukaryotic community composition at all sites was dominated by Chloroplastida and fungi, which is consistent with the findings of previous research conducted in Svalbard [39]. Chloroplastida play an important role in the formation and function of biocrusts due to their photosynthetic capabilities. Furthermore, the relatively high abundance of bryophytes within the eukaryotic groups is characteristic of Arctic tundra in this region [1]. The bacterial community in biocrusts was dominated by Proteobacteria and Actinobacteria across all study sites. While minor variations in community composition were observed between sites, the overall ratio of bacterial phyla remained relatively stable, aligning with the findings from studies of biocrusts under diverse environmental conditions [2,6].

Photoautotrophic species, including green algae, lichens, bryophytes and cyanobacteria, are frequently observed to be abundant in biocrusts [4]. The biocrusts exhibited a high diversity of green algae from families Trebouxiophyceae, Chlorophyceae and Chrysophyceae being the most prominent among the eukaryotic algae. While the majority of green algal classes demonstrated a decline in abundance with increasing altitude, Trebouxiophyceae showed the opposite trend, becoming more prevalent at higher altitudes (except Site 2). This pattern is likely driven by the well-documented symbiotic associations between these taxa and lichens [2,8], linking their increased abundance to the presence of lichen species. At Site 4 (314 m a.s.l.), where lichen-dominated biocrusts are prominent, this trend was further supported by the high abundance of Proteobacteria and cyanobacteria, which are commonly associated with lichen-rich environments [40]. A further analysis of fungal guilds highlighted the prevalence of lichen-dominated biocrusts at Sites 2 and 4, where lichenised fungi reached their highest abundance. The Ascomycota fungus *Verrucaria*, known for its symbiotic relationships with green algae from the *Trebouxiophyceae* and *Ulvophyceae* taxa [41], was the most abundant overall. Trebouxiophyceae, in particular, exhibited higher abundances at Site 4. Additionally, *Umbilicaria*, a lichen species frequently found on rocks at higher elevations in Arctic climates [42,43], showed an increased number of reads, particularly at Site 3. The photobiont of this lichen, the green algae *Trebouxia* sp. [42], exhibited a similar trend, thereby supporting the known ecological association between them.

The photoautotrophic community composition at all sites was dominated by cyanobacteria, which is consistent with the previous research that emphasised their role as early colonisers, soil stabilisers and integral components of biocrusts [2,4]. In contrast, bryophytes and eukaryotic green algae are typically associated with later stages of biocrust development. This indicates that Sites 1 and 3 may represent more advanced stages of biocrust formation, as they exhibited the highest abundance of these later successional groups [2,5].

The cyanobacterial community was dominated by the orders Pseudoanabaenales, Gloeobacterales and Nostocales. Filamentous species such as *Leptolyngbya* sp., identified as an indicator species at Sites 2 and 4, play a pivotal role in soil stabilisation. By secreting sticky extracellular polymeric substances (EPS), they facilitate soil particle aggregation, forming a foundation for biocrust development [3,44]. These cyanobacteria are often dominant within biocrusts and contribute significantly to their formation and stability [45]. The order Nostocales, which includes heterocystous species such as *Nostoc* sp., was relatively abundant across all four sites. Heterocystous cyanobacteria are capable of fixing atmospheric nitrogen, thereby supplying essential nutrients to other soil organisms [6,46]. *Nostoc* sp., frequently observed in Arctic biocrusts, contributes significantly to primary production [1] and thrives in habitats with elevated moisture levels [47]. This was reflected in its high abundance at Site 4, where the highest soil moisture was recorded. Previous studies have demonstrated that while the diversity of cyanobacteria is not affected by altitude, their overall abundance tends to increase at higher altitudes [48,49]. Cyanobacteria are well adapted to harsh conditions due to their rapid metabolic rates, resilience to water shortages, and ability to withstand extreme temperature fluctuations [46,48]. These findings are consistent with our observations, as cyanobacterial community composition showed little variation between sites, but their overall abundance increased with altitude. Under favourable environmental conditions, heterocystous cyanobacteria play a crucial role in fixing atmospheric nitrogen, which not only supports the growth of other soil organisms but may also benefit lichen species [46]. Recent studies have highlighted the mutualistic relationships between microalgae, nitrogen-fixing bacteria, and lichens, forming so-called tripartite lichens [50], suggesting that this interaction may enhance the stability and functionality of Arctic biocrusts [50,51]. Consequently, the high abundance of *Nostoc* sp. at Site 4 may contribute to primary production while providing essential nutrients for the development of lichen communities, highlighting the complex ecological interactions within biocrust ecosystems.

Overall, the observed changes in photoautotrophic communities appear to offset each other, as no significant changes in total chlorophyll *a* concentration were observed at the different sites. This suggests that the rate of primary production was consistent across all sites. Similarly, no significant differences were detected in the composition of the photoautotrophic community, indicating a stable functional contribution of these organisms to primary production under the varying environmental conditions studied. In accordance with this, measurements of photosynthetic capacity using a field PAM revealed no significant differences between the sites. These findings are consistent with other studies on biological soil crusts [52], which reported that variations in photoautotroph abundance were not directly correlated with factors such as crust age or the activity of phototrophic microorganisms. Such resilience underscores the adaptive capacity of photoautotrophic organisms in maintaining consistent primary production even under challenging and variable environmental conditions.

However, these findings are subject to certain limitations. As the study focused on a single toposequence, site-specific differences in physicochemical parameters may influence the results. Additionally, seasonal aspects were not considered, limiting the scope of these data to summer conditions. Potential disturbances, such as freeze-thaw cycles or animal activity, which could affect soil and biocrust dynamics, were also not monitored.

While the overall community composition did not show a clear correlation with the altitudinal gradient, the abundance of different phyla reflected the environmental conditions at each site, which were influenced by the altitudes. Previous studies demonstrated that bacterial composition is particularly sensitive to pH, with neutral soils supporting higher bacterial diversity [6,53]. The highest bacterial diversity was observed at Sites 2 and 4, both characterised by pH values close to 7. Furthermore, lower pH values were associated with a higher abundance of Acidobacteria, whereas Proteobacteria were more abundant in near-neutral soils [54]. This pattern aligns with the results observed at Sites 2 and 4, where Proteobacteria were the dominant phylum and pH levels were close to 7. The NMDS analysis further confirmed that environmental factors, including temperature, moisture, pH, nitrogen and phosphorus content, significantly influenced the community composition.

## 5. Conclusions

Biocrusts are essential to Arctic tundra ecosystems, providing a habitat for a diverse range of organisms and contributing to key ecological processes. Despite their importance, a complete understanding of the interactions among species within biocrusts remains a challenge. This study focused on the community composition of photoautotrophic organisms, with a particular emphasis on cyanobacteria and eukaryotic photoautotrophs. While elevation was not identified as a primary factor influencing microbial communities, environmental conditions played a significant role in shaping community variation. The bacterial community was primarily influenced by shifts in pH, whereas the abundance of eukaryotic organisms, including vascular plants, was more responsive to broader environmental factors. These findings highlight the complexity of Arctic biocrust micro-ecosystems and the significant influence of specific environmental conditions on their structure.

## Figures and Tables

**Figure 1 microorganisms-12-02606-f001:**
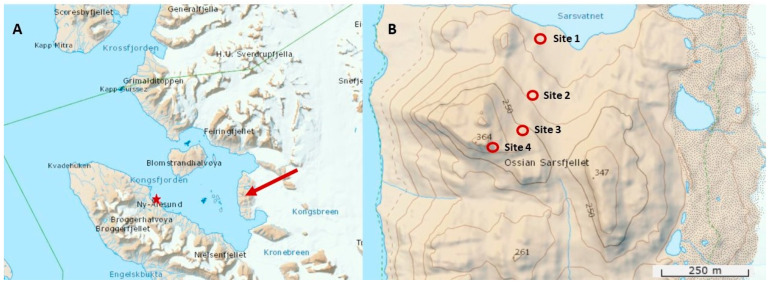
Geographical location of sampling sites on Ossian Sarsfjellet, Svalbard. (**A**)—The red arrow indicates the location of Ossian Sarsfjellet and the red star indicates the location of the weather station in Ny-Ålesund in the Kongsfjorden area. (**B**)—The red circles indicate the sampling sites. Maps based on TopoSvalbard, courtesy of the Norwegian Polar Institute.

**Figure 2 microorganisms-12-02606-f002:**
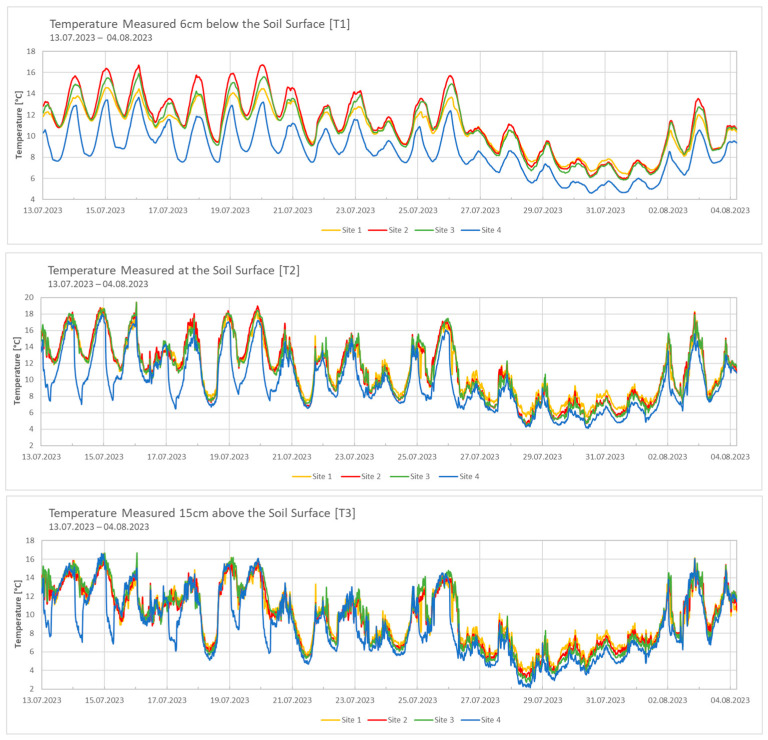
Temperature data from 13 July 2023 to 4 August 2023 at the three different measurement points across the four sites. T1 = measured 6 cm below the soil surface; T2 = measured at the soil surface; T3 = measured 15 cm above the soil surface; measured every 15 min; measured with TOMST^®^ data loggers.

**Figure 3 microorganisms-12-02606-f003:**
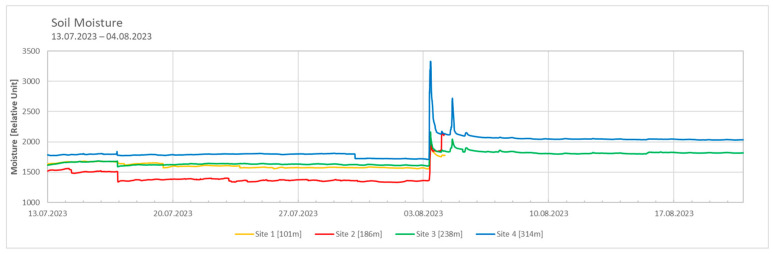
Soil moisture data from 13 July 2023 to 20 August 2023 across the sites. Measured every 15 min; measured with TOMST^®^ data loggers; significant differences observed between all four sites; *p*-value < 0.05. Data used for statistical analysis are from 13 July 2023 to 4 August 2023 only.

**Figure 4 microorganisms-12-02606-f004:**
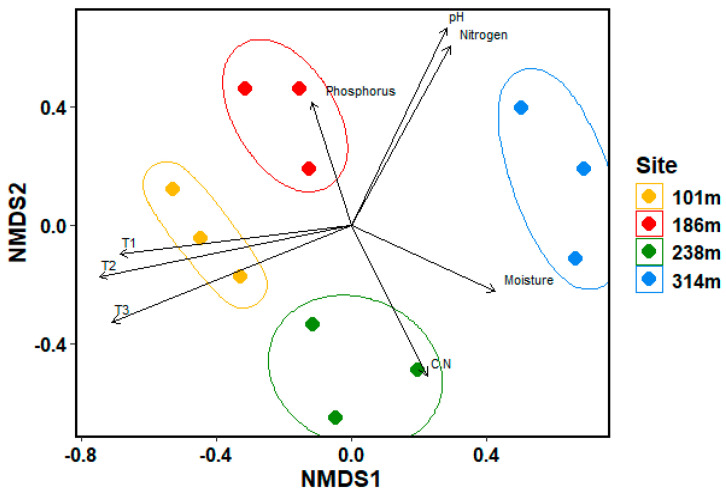
Non-Metric Multidimensional Scaling (NMDS) plot based on the vegetation analysis conducted at the sites. T1 = temperature measured 6 cm below the soil surface; T2 = temperature measured at the soil surface; T3 = temperature measured 15 cm above the soil surface; C.N = carbon to nitrogen ratio; parameters which did not show significant differences between the sites are not included, *p*-value < 0.05.

**Figure 5 microorganisms-12-02606-f005:**
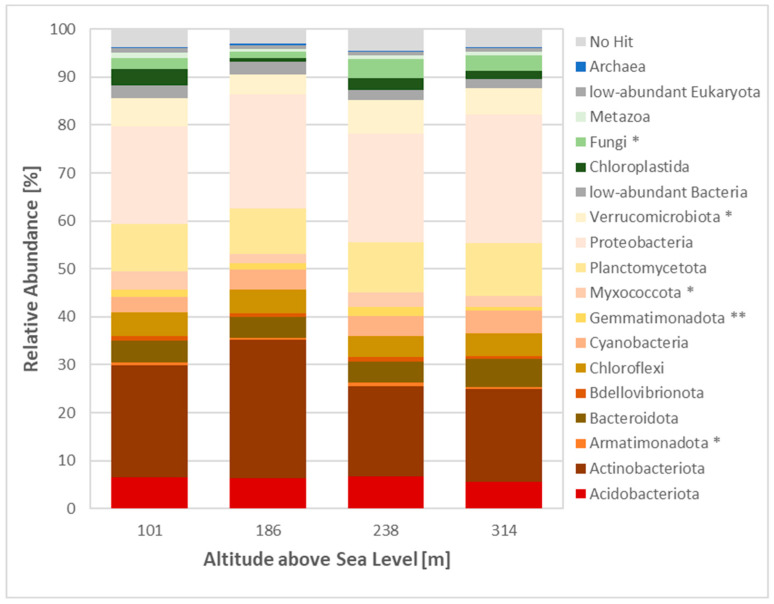
Overall community composition across sites. Metagenomic 16/18 S rRNA dataset analysed using Silva software; taxonomic groups with total abundance below the 0.5% threshold are grouped as ‘low abundance’; * indicates significant differences between sites based on a one-way ANOVA, *p*-value * < 0.05, ** < 0.01).

**Figure 6 microorganisms-12-02606-f006:**
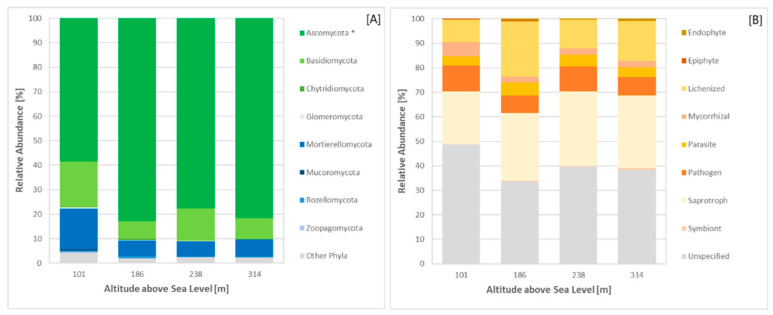
Relative abundance of fungal taxa and fungal functional guilds across sites. Metagenomic 16/18S rRNA dataset analysed using Silva software. (**A**)—fungal phyla. (**B**)—functional guilds; * indicates significant differences between sites based on a one-way ANOVA, *p*-value < 0.05.

**Figure 7 microorganisms-12-02606-f007:**
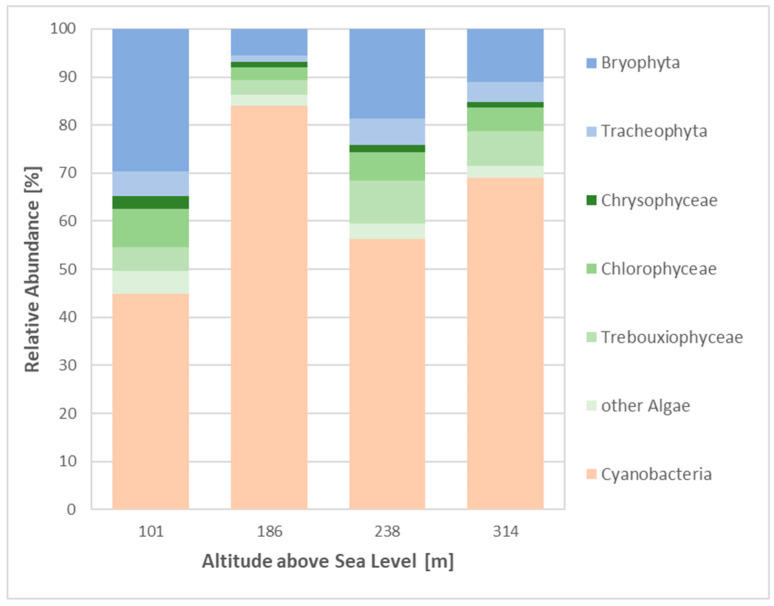
Relative abundance of photoautotrophic taxa across sites. Metagenomic 16/18S rRNA dataset analysed using Silva software; no significant differences were observed between the taxa, based on a one-way ANOVA, *p*-value < 0.05.

**Figure 8 microorganisms-12-02606-f008:**
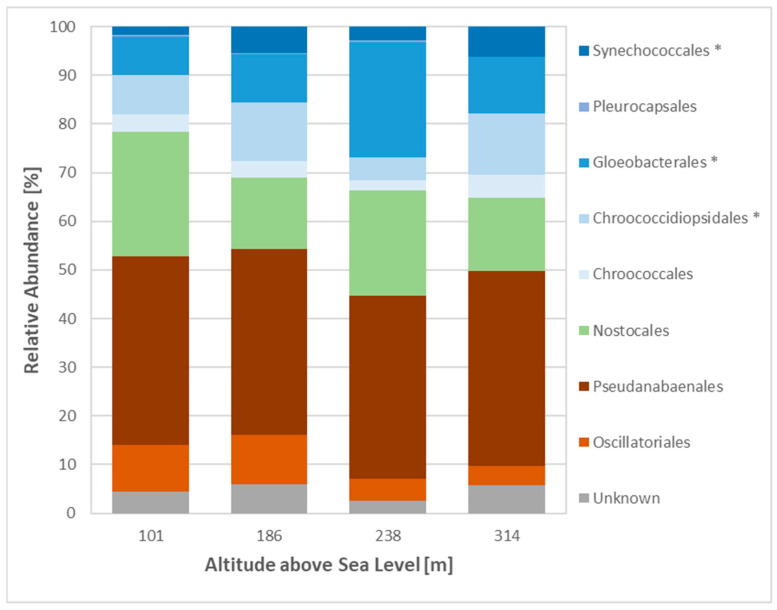
Relative abundance of cyanobacterial orders across sites. Metagenomic 16 S rRNA dataset analysed using Silva software; filamentous orders shown in orange/red; heterocystous order shown in green; unicellular orders shown in blue; * indicates significant differences between sites based on a one-way ANOVA, *p*-value < 0.05.

**Table 1 microorganisms-12-02606-t001:** Scale used for the determination of soil coverage ratio of organisms on the sampling sites. Categorisation according to visual assessment.

Scale	Coverage	Numeric Conversion for Statistical Purposes
5	>75% of surface	Did not occur
4	51–75% of surface	Did not occur
3	26–50% of surface	50
2	5–25% of surface	25
1	<5% of surface, but many individuals	10
+	<5% of surface, but few individuals	5
r	rare	1

**Table 2 microorganisms-12-02606-t002:** Description of the studied sites, including soil chemistry. The same letters indicate no statistical difference between the samples according to a one-way ANOVA, followed by the Tukey HSD post hoc test (*p* < 0.05); TP = total phosphorus; TN = total nitrogen; TC = total carbon; C/N = carbon to nitrogen ratio; Chl *a* = chlorophyll *a* content; Fv/Fm = quantum yield capacity of photosystem II.

Site	GPS	Elevation a.s.l. [m]	pH	TP, [g/kg]	TN, [g/kg]	TC, [g/kg]	C/N	Chl *a*, [mg/m^2^]	Fv/Fm
1	78.94935° N 12.48538° E	101	6.26 ^a^	0.42 ^a^	4.67 ^ab^	73.30 ^a^	15.77 ^ab^	183.88 ^a^	0.343 ^a^
2	78.94562° N 12.48588° E	186	7.28 ^b^	0.39 ^a^	5.53 ^a^	76.33 ^a^	13.79 ^a^	151.39 ^a^	0.305 ^a^
3	78.94352° N 12.48400° E	238	6.13 ^a^	0.27 ^a^	3.40 ^b^	64.80 ^a^	18.66 ^b^	137.33 ^a^	0.353 ^a^
4	78.94226° N 12.47245° E	314	7.13 ^b^	0.37 ^a^	5.97 ^a^	99.60 ^a^	16.61 ^ab^	215.57 ^a^	0.344 ^a^

## Data Availability

All sequence reads were submitted to the NCBI database project number PRJNA1189472.

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
