# Peer review of "Role of Climate and Edaphic Factors on the Community Composition of Biocrusts Along an Elevation Gradient in the High Arctic"

_microorganisms, 2024, doi:10.3390/microorganisms12122606_

Round 1
Reviewer 1 Report
Comments and Suggestions for Authors
The manuscript entitled ‘’Role of climate and edaphic factors on the community composition of biocrusts along an elevation gradient in the High Arctic’ is to enhance the understanding of the microbial community composition of Arctic biocrusts and the climatic and edaphic factors influencing these communities. The work is very interesting. But there are something need to be improved.
1. The number of the introduction is too many. I suggested the author decrease the number of paragraphs.
2. In the 2.4 of method and materials, the author should add more details of the DNA and following experimental processes.
3. In general, the sample collection in a short period. It is hard to conclude the influence on the community composition by climate. Insofar as I can see, a long period observation is necessary if we are interested in the effects of climate. So, I suggested the author changed the title.
4. Overall, the author gave a highlight of the microbial community composition. However, the author only showed more information about algae (as shown in figure 6 and figure 7). I suggested the author would add more information about fungi or other information in details other than in the supplementary. Especially, some relevant figures would be better placing in the result and discussion.
5. I suggested the author could add some analysis aiming at the environmental factors on microbial function. Because, the conclusion displayed that the ‘ecosystems and the significant influence of specific environmental conditions on their structure and function’
Comments on the Quality of English Languageno comments
Reviewer 2 Report
Comments and Suggestions for Authors
1. The manuscript is well-written, but minor grammatical improvements and sentence restructuring could enhance readability. For example, sentences in the introduction are dense and could benefit from simplification.
2. The abstract is comprehensive but could emphasize the ecological implications of the findings more prominently to enhance reader engagement.
3. Introduction: Expand on the novelty of the study compared to prior research in the Arctic region.
4. Introduction: Consider adding more details about why the selected elevation gradient (106–306 m) is particularly relevant.
5. While the figures effectively support the results, their resolution in the manuscript appears low. High-resolution versions with clearer labels are recommended.
6. Broaden the discussion on the ecological implications of microbial dynamics under changing Arctic conditions.
7. Address limitations, such as potential seasonal variability in microbial communities or the influence of undetected factors.
Reviewer 3 Report
Comments and Suggestions for Authors
Dear Authors,
I think the article is well-written. However, experimentally it has few results. Therefore, an important part should be a thorough discussion of the existing literature, and in my view, this is the area where the authors should make the strongest improvements.
-Explain the rationale behind the selection of sampling sites and the study period
-L77: “2.1 Site description and sampling” I would advise the authors to include the Cartesian coordinates
-Detail the metagenomic analysis process further, including the specific parameters used in the OmicsBox software
-Enhance the resolution and clarity of Figure 1, particularly the map of the sampling sites
-L207: “dew formation of dew” Improve the grammatical expression
-L383: “This variation suggests that additional variables, such as subtle shifts in community composition, geological conditions, plant-microbe interactions or specific soil characteristics, may play an important role in shaping the nutrient profile at this site.” Is there any supporting literature for this hypothesis? Please include it
-L389-L91: Please a reference is needed
-L401: “(reference).” I agreed too
-L411: “Prestø's”? I guess reference 34
-L420: “proved to be challenging” Please discuss the reasons with supporting the literatura.
-L465: “cyanobacteria are capable of fixing atmospheric nitrogen…”In relation to this and the amount of lichens found in each site, it has been recently described the interaction between microalgal and nitrogen-fixing bacterial consortia, and its relationship with Lichens. Could this have an implication with the result obtained ? Please discuss
-L474: “extreme temperature fluctuations” and the ability to fix nitrogen too? Please discuss in more detail
-L477-481: Please try to support this hypothesis with the related bibliography
Round 2
Reviewer 3 Report
Comments and Suggestions for Authors
I believe the authors have adequately addressed all of my comments and suggestions, and I accept the paper in its current version.